# Combination Regimens with Colistin Sulfate versus Colistin Sulfate Monotherapy in the Treatment of Infections Caused by Carbapenem-Resistant Gram-Negative Bacilli

**DOI:** 10.3390/antibiotics11101440

**Published:** 2022-10-19

**Authors:** Min Hao, Yang Yang, Yan Guo, Shi Wu, Fupin Hu, Xiaohua Qin

**Affiliations:** 1Institute of Antibiotics, Huashan Hospital Fudan University, Shanghai 200040, China; 2Key Laboratory of Clinical Pharmacology of Antibiotics, National Health Commission, Shanghai 200040, China; 3Shanghai Huashen Institute of Microbes and Infections, Shanghai 200052, China

**Keywords:** carbapenem-resistant Gram-negative bacilli, colistin sulfate, combined therapy, clinical efficacy, safety evaluation

## Abstract

Carbapenem-resistant organisms (CRO) have become a global concern because of the limited antibiotic treatment options for CRO infections. Colistin sulfate is a type of polymyxin approved for the treatment of CRO in China. To date, studies on polymyxin have mainly focused on in vitro antibacterial activity or pharmacokinetics/pharmacodynamics, and few have evaluated its clinical efficacy. We aimed to compare the clinical efficacy and safety of colistin sulfate monotherapy and its combination with other antimicrobials in the treatment of carbapenem-resistant Gram-negative bacilli (CR-GNB) infections in adults. This retrospective study included adult patients with CR-GNB infections treated with colistin sulfate by intravenous drip between January and June 2020. The patients were divided into two groups, according to the administration of colistin sulfate alone or in combination with other antibiotics. Group-wise demographic data, comorbidities, clinical efficacy, prognosis, and adverse events were analyzed and compared. In total, 26 patients in the colistin sulfate monotherapy group and 54 patients in the combined therapy group were recruited. The clinical efficacy in the combined therapy group (94.4%) was significantly higher than that in the colistin monotherapy group (73.1%) (*p* = 0.007); however, the 28-day mortality and length of hospital stay were not significantly different between groups. The incidence of adverse events (including elevated aminotransferase, bilirubin, serum creatinine, and decreased platelet) was not significantly different between the groups. Combination therapies with colistin sulfate are recommended for the treatment of CR-GNB infections, over colistin sulfate alone.

## 1. Introduction

Carbapenem-resistant organisms (CRO) have become a global concern because of the limited antibiotic treatment options for CRO infections, particularly carbapenem-resistant *Klebsiella pneumoniae* (CRKP), carbapenem-resistant *Acinetobacter baumannii* (CRAB), and carbapenem-resistant *Pseudomonas aeruginosa* (CRPA). Polymyxin, a polypeptide antibiotic, developed in 1947 to treat Gram-negative bacterial infections, has gained renewed attention as a first-line drug for treating CRO infections. Due to its poor oral absorption, polymyxin is always administered intravenously. However, the clinical use of polymyxin has been restricted owing to nephrotoxicity. Several reviews and guidelines have recommended a combination regimen containing tigecycline, polymyxin, ceftazidime, and avibactam for treating multidrug-resistant Gram-negative bacterial infections [1,2].

Colistin and polymyxin B are two common types of polymyxin [3]. Polymyxin B is recommended for the treatment of serious infections, such as bloodstream infections. In contrast, colistin is recommended for urinary tract infections, because its concentration in the urinary tract is higher than that of polymyxin B [4]. However, prospective clinical studies are warranted, to evaluate the differences in the efficacy and safety of the two agents in different types of infections. Colistins are categorized as cationic polypeptide antibiotics. Two forms of colistin are commercially available worldwide; namely, colistin sulfate and the sodium salt of the negatively charged derivative of colistin (known as colistin methanesulfonate (CMS)) [3]. Unlike colistin sulfate, which is active in vivo, colistin methanesulfonate is an inactive prodrug in vivo. CMS-Na is administered intravenously and CMS, the free base of CMS-Na consisting of CMS A and CMS B, is hydrolyzed in vivo to form active colistin A and colistin B6 [5,6].

To date, studies on polymyxin have mainly focused on in vitro antibacterial activity or pharmacokinetics/pharmacodynamics [7,8,9], while some have evaluated the clinical and microbiological efficacy of polymyxin B [10,11]; however, few have evaluated the clinical efficacy of colistin sulfate [12]. Colistin sulfate has only been listed in China since 2018 and is approved for blood, urinary tract, and pulmonary infections caused by CRO. Here, we conducted a retrospective study on the efficacy of colistin sulfate monotherapy and combination therapy on carbapenem-resistant Gram-negative bacilli (CR-GNB), to provide a reference for the clinical use of colistin sulfate.

## 2. Materials and Methods

### 2.1. Ethical Approval of the Study Protocol

The study protocol was approved by the Ethics Committee of the Fudan University Huashan Hospital (KY2019-460), Shanghai, China. The study was conducted according to the ethical standards of the 1964 Declaration of Helsinki and its later amendments. Informed consent was not required in this study because of its retrospective nature.

### 2.2. Patients

This retrospective study enrolled patients admitted to the hospital from 1 January to 31 August 2020. Forty-nine hospitals (all members of CHINET, the China Antimicrobial Surveillance Network) from 13 provinces in China participated in this study. Eighty-nine cases reported the usage of colistin sulfate during the study period and finally 80 cases were included (Figure 1). The inclusion criteria were as follows: (a) intravenous administration of colistin sulfate (dose of 1–20 million U/kg); (b) age ≥ 18 years; (c) diagnosis of hospital-acquired pneumoniae, blood infection, urinary tract infection, acute suppurative peritonitis, or acute suppurative meningitis; and (d) isolated CR-GNB from lower respiratory tract secretions, blood, mid-stream urine, abdominal drainage, cerebrospinal fluid, or other specimens. The exclusion criteria were as follows: (a) CR-GNB considered as colonization strains according to the history; (b) infections caused by combined Gram-positive coccus; (c) colistin sulfate and other antimicrobial agent administration <72 h after CR-GNB isolation; and (d) kidney dysfunction (acute renal damage, chronic kidney disease stage 3 or more) or receiving renal replacement therapy.

### 2.3. Data Collection

Patients treated with intravenous colistin sulfate only (Shanghai Xinya Pharmaceutical Co., Shanghai, China) were assigned to the monotherapy group, and those treated with intravenous colistin sulfate and other antimicrobial agents were assigned to the combination therapy group.

Demographic characteristics, including age, sex, underlying disease, comorbidities, type of infection, bacterial susceptibility to antibacterial agents, treatment duration, combination therapy regimen, body temperature, 7-day clinical inflammation index, length of hospitalization, and mortality rate at 28 days and discharge, were recorded.

### 2.4. Clinical Efficiency Evaluation

Clinical efficacy % = (number of cured cases + number of improved cases)/total number of cases × 100%. The observation time of antimicrobial treatment efficacy was defined as within 24 h of the discontinuation of the therapeutic drugs. The clinical outcomes were divided into three conditions (cure, improvement, and failure). A cure was defined as the complete disappearance of clinical signs and symptoms of infection, with normal total white blood cell count, neutrophil ratio, C-reactive protein (CRP), procalcitonin (PCT), and radiographic and pathogenic examinations. Improvement was defined as not fully achieving the cure index, with at least two of the clinical symptoms and signs, the previously mentioned inflammatory indices, or radiographic and pathogenic examination results reaching an improvement of >50% compared with pre-treatment, and no observed deterioration for the rest of the indices. Treatment failure was defined as no improvement in or aggravation of clinical symptoms and signs, no significant decrease or an increase in inflammatory indices, radiographic findings suggesting the progression of infectious lesions, microbiological examination of the original bacteria consistently isolated from clinical specimens, or patient death before efficacy assessment. Microbiological clearance efficiency was defined as the disappearance or clearance of CR-GNB from clinical specimen cultures after treatment [13].

A decrease in inflammatory indices was defined as a >20% difference in total white blood cell count, neutrophil ratio, CRP, or PCT between ±2 days and 7 ± 2 days from the treatment initiation date. The prognosis was assessed based on the 28-day mortality rate and the mortality rate at hospital discharge.

### 2.5. Safety Evaluation

Relevant laboratory indicators were collected within 24 h of treatment discontinuation, to evaluate the safety of colistin sulfate. Renal impairment was defined as a blood creatinine level above normal (17.7–107 μmol/L) or double that of the baseline. Liver function impairment was defined as alanine aminotransferase (≤37 U/L), aspartate aminotransferase (≤37 U/L), and total bilirubin (≤25 μmol/L) levels above normal or double those at baseline. Thrombocytopenia was defined as a platelet decline to ≤100 × 10^9^/L during colistin sulfate administration (except for platelet decline caused by primary hematologic disease). The baseline was defined as the result reported ±24 h from the initiation of colistin sulfate administration.

### 2.6. Microbiology

CR-GNB strains were isolated from body fluid specimens of enrolled patients (including lower respiratory tract secretions, blood, mid-stream urine, abdominal drainage, and cerebrospinal fluid). In vitro antimicrobial susceptibility testing was performed according to the Clinical and Laboratory Standards Institute (CLSI) guidelines (M100), using the micro-broth dilution method [14]. The results for piperacillin/tazobactam, ceftazidime, cefoperazone/sulbactam, cefepime, imipenem, meropenem, ciprofloxacin, levofloxacin, amikacin, minocycline, doxycycline, sulfamethoxazole/trimethoprim, and tigecycline were interpreted according to CLSI breakpoints [14]. The colistin resistance level was interpreted according to the United States Committee on Antimicrobial Susceptibility Testing [15]. “Carbapenem resistance” was defined as a minimal inhibitory concentration (MIC) of imipenem or meropenem ≥4 mg/L or ertapenem ≥2 mg/L.

### 2.7. Statistical Analysis

Statistical analysis was performed using Stata SE15.0. Continuous variables are presented as mean  ±  standard deviation if normally distributed and as median M (first quartile Q1, third quartile Q3) if not normally distributed. Continuous variables were compared using a *t*-test and chi-square test if they conformed to normality; if not, the rank-sum test was used. A chi-square or Fisher’s exact probability test was performed to compare categorical variables. Statistical significance was set at *p* < 0.05.

## 3. Results

### 3.1. Participant Characteristics

Eighty patients treated with colistin sulfate with or without intravenous antimicrobial agents were included in this study. The typical dosage of colistin sulfate administered to 74 patients was 0.5 million U every 12 h (1 million U for the first dose). The dosage administered to six patients was 1.5 million U per day, because of their high weight (to three patients, it was 0.5 million U every 8 h, 1 million U for the first dose, to another three patients, it was 0.75 million U every 12 h, 1.5 million U for the first dose). This high-weight regimen was only used in the monotherapy group. Combination regimens with tigecycline, carbapenems, cefoperazone/sulbactam, piperacillin/tazobactam, ceftazidime/avibactam, aminoglycosides, and levofloxacin were included. Imipenem (1.0 g) every 8 h or meropenem (1.0 g) every 8 h was the most common dosage of carbapenems. Meropenem (2.0 g) was administered every 8 h in three cases of central nervous system infection. The normal dosage of tigecycline was 50 mg every 12 h (100 mg for the first dose), and 100 mg every 12 h (200 mg for the first dose) was used in four cases. The dosage of cefoperazone/sulbactam was 3.0 g every 6 h, piperacillin/tazobactam was 4.5 g every 8 h, amikacin was 0.4 g per day, and levofloxacin was 0.5 g per day.

No significant differences were observed in age, sex, underlying disease, invasive manipulation 1 week before infection, combined use of glucocorticoids or immunosuppressants, site of infection, or distribution of CR-GNB between the colistin monotherapy and combination therapy groups (Table 1 and Table 2).

Most patients were admitted to the ICU (66.3%). The non-ICU group cases were mainly from the respiratory (7.5%), hematology (7.5%), transplantation (5%), and neurosurgery (5%) departments.

### 3.2. Clinical Efficiency Evaluation

Patients in both the monotherapy and combination groups had many underlying diseases and long hospitalization days. The mean number of hospitalization days in the monotherapy and combination groups was 47.2 ± 32.4 and 49.8 ± 35.0, respectively, with no significant difference (*p* > 0.05). Among patients with bloodstream infection, the duration of treatment was significantly longer in the combination therapy group (14.5 ± 2.1 days) than in the monotherapy group (9.7 ± 4.3 days) (*p* = 0.011).

The number of patients whose WBC counts returned to normal (72.6% vs. 42.3%), and the neutrophilic granulocyte percentage (70.6% vs. 38.5%), CRP (92.6% vs. 57.7%), and PCT decline (87% vs. 53.9%), within 7 days were significantly higher in the combination therapy group than in the monotherapy group (*p* ≤ 0.05) (Table 3). Moreover, the clinical efficacy (94.4% vs. 50.1%, *p* = 0.007) and microbiological clearance efficiency rate (74.1% vs. 50%, *p* = 0.033) were significantly higher in the combination therapy group than in the monotherapy group. No significant differences were observed between groups in terms of treatment duration, hospitalization days, time to normalization of body temperature, improvement of inflammatory indices, clinical efficiency index, 28-day mortality, and discharge mortality (Table 3).

The distribution of CR-GNB was different among the combination therapy regimens (*p* = 0.013). *A. baumannii* was mostly treated with colistin combined with tigecycline (66.7%). *P. aeruginosa* was treated with colistin combined with cefoperazone/sulbactam (42.9%) or aminoglycosides (66.7%). Colistin combined with carbapenem and tigecycline was mainly used to treat *A. baumannii* (42.9%) and *K. pneumoniae* (42.9%) infections.

### 3.3. Safety Evaluation

No significant differences were observed in the incidence of hepatic impairment, renal impairment, or thrombocytopenia between the colistin sulfate monotherapy and combination therapy groups (Table 4). All five patients in the monotherapy group treated with colistin sulfate at a daily dose of 1.5 million U developed renal impairment during administration.

### 3.4. Susceptibility Testing

Antimicrobial susceptibility testing was performed on 50 CR-GNB strains. Eleven of the 50 strains were *K. pneumoniae*, and all were resistant to piperacillin/tazobactam, ceftazidime, cefoperazone/sulbactam, cefepime, imipenem, and meropenem; only 36% were resistant to amikacin. Moreover, 100% of *K. pneumoniae* strains were sensitive to tigecycline and colistin. The MIC_50_ for tigecycline was 2 mg/L and that for colistin was ≤0.5 mg/L. Thirteen of the 50 strains were *P. aeruginosa*, and 100% were resistant to piperacillin/tazobactam, cefoperazone/sulbactam, imipenem, and meropenem, whereas 69% were resistant to ceftazidime, 77% to cefepime, and 31% to amikacin. All *P. aeruginosa* strains were sensitive to colistin. Twenty-three of the 50 strains were *A. baumannii*, and 43% were resistant to minocycline, 9% to tigecycline, and 0% to colistin. The resistance rates of *A. baumannii* to other antimicrobial agents were >90%. Three of the 50 strains were *S. maltophilia*, and all were sensitive to levofloxacin, minocycline, tigecycline, and colistin (Table 5).

## 4. Discussion

In recent years, polymyxins have become one of the most important treatment options for extensively drug-resistant Gram-negative bacterial infections. Colistin methanesulfonate and polymyxin B sulfate, two common types of polymyxin [3], are two common agents available abroad. They are currently available in China. This study analyzed cases of confirmed CR-GNB infections treated with intravenous colistin sulfate in China since its launch.

Combination therapy is recommended for treating CR-GNB infections [2,16,17,18,19]. This study included colistin sulfate monotherapy and combination regimens with tigecycline, carbapenems, cefoperazone/sulbactam, piperacillin/tazobactam, ceftazidime/avibactam, aminoglycosides, and levofloxacin for the treatment of CR-GNB infections. Most dosing regimens in this study were according to the guidelines for extensively drug-resistant Gram-negative infections, such as colistin in combination with tigecycline for the treatment of CRAB infection [18] and in combination with ceftazidime/avibactam for the treatment of CRPA [20] and CRKP [21] infections. Guidelines recommend colistin in combination with carbapenems for isolates with an MIC ≤8 mg/L for carbapenems, and a high loading dose and prolonged duration are needed [2]. In this study, the MICs for carbapenems were ≥16 mg/L, and a regular dose was used. However, no significant difference in clinical efficacy was observed between the carbapenem and tigecycline combination groups, which might have resulted from the small sample size or lack of severity stratification.

Considering severe nephrotoxicity, colistin combined with aminoglycosides is not recommended. In this study, two patients were treated with this regimen. However, the blood creatinine level only increased onefold from baseline, but was still within the normal range according to their medical history. This might have been related to the small dose of amikacin used (0.4 g per day).

In this study, no significant differences were observed with age, sex, underlying disease, risk factors for CR-GNB infection, type of infection, pathogen distribution, time from positive culture to the start of anti-infective treatment, or duration of treatment between the colistin sulfate monotherapy and combination therapy groups. However, the improvement rate of clinical inflammatory indices, change in WBC count, percentage of neutrophils, and PCT, and clinical efficiency and microbial clearance rates were significantly higher in the combination therapy group than in the monotherapy group 7 days after initiating treatment. Abdelsalam et al. reported [22] that the clinical efficacy of colistin methanesulfonate combined with meropenem in treating multiple drug-resistant *K. pneumoniae* in hospital-acquired ventilator-associated pneumoniae is 83.3% (25/30), which is significantly higher than that in the monotherapy group (56.7%, 17/30). No significant difference was observed in the incidence of acute renal impairment between the groups [22]. However, a randomized controlled superiority trial [23] reported that the addition of meropenem to colistin does not improve clinical failure in severe *A baumannii* infections.

The MICs for various antimicrobial agents against 50 CR-GNB strains were determined in this study. The resistance rate of the 23 *A. baumannii* isolates to amikacin was >90%, whereas that of minocycline was <50%. Only oral formulations of minocycline are available in China, and they are not recommended for the treatment of severe infections. However, minocycline can be used in combination with other drugs to treat mild CRAB infections. The resistance rates of 13 *P. aeruginosa* isolates and 11 *K. pneumoniae* isolates to amikacin were >40% (MIC_50_ ≤ 4 mg/L), which was consistent with the results of CHINET [24]; therefore, amikacin is a wise choice for the combination treatment of CRKP and CRPA [2]. Tigecycline and colistin have become the mainstay treatments for CR-GNB infections in recent years [16,17,18,19]. *P. aeruginosa* was naturally resistant to tigecycline, whereas the resistance rates of *A. baumannii* and *K. pneumoniae* to tigecycline were less than 10% (MIC_50_ ≤ 1 mg/L). All CRAB, CAKP, and CRPA strains were susceptible to colistin (MIC_50_ ≤ 0.5 mg/L).

This study has several limitations. First, selection bias may have been inevitable because of the small sample size. In this retrospective case-cohort study, the colistin sulfate dose in each case was not completely consistent. Therefore, large-scale randomized controlled studies are required to further assess the efficacy of colistin sulfate. Second, the cases were not stratified based on the severity of the infections. Third, medical histories related to adverse reactions were lacking; therefore, no recommendation on colistin dose could be given from the perspective of adverse reactions. Nonetheless, our study provides a basis for the optimal selection of combination regimens against CR-GNB infections based on the antimicrobial resistance of pathogenic bacteria.

## 5. Conclusions

In summary, combination regimens, including colistin sulfate, are recommended for treating CR-GNB infection. The clinical efficacy and microbiological clearance efficiency were better in the combination therapy group than in the monotherapy group. The resistance rates of CR-GNB to antimicrobial agents varied. The combination of antimicrobial agents should be based on the resistance characteristics of different pathogenic bacteria. Our findings provide evidence for the clinical efficacy and safety of colistin based on clinical application and may help clinicians better select a treatment regimen for CRO infections. However, further clinical studies are required, to clarify the impact of combination therapy versus monotherapy on the long-term prognosis and benefits.

## Figures and Tables

**Figure 1 antibiotics-11-01440-f001:**
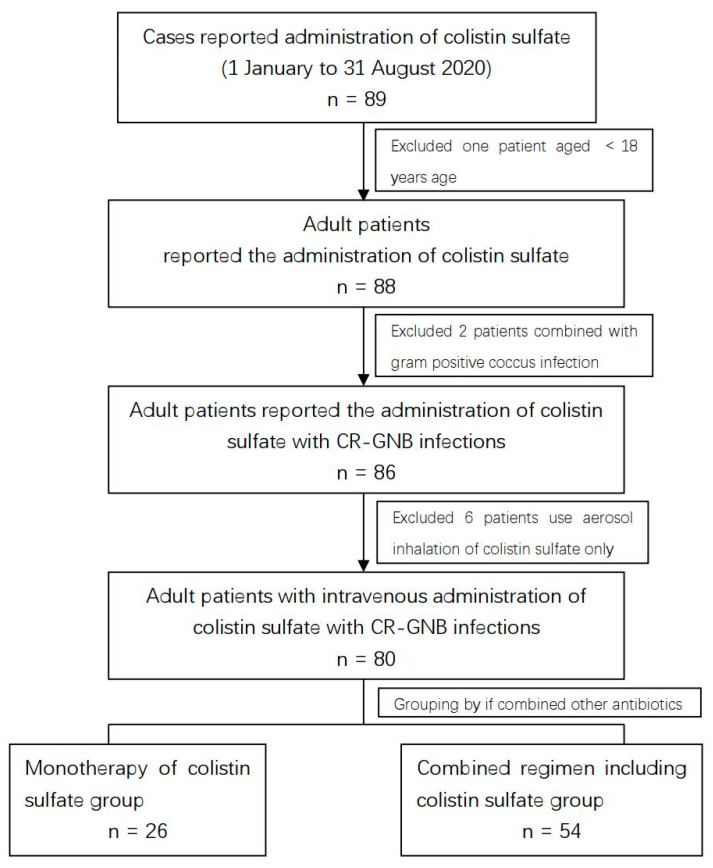
Case screening flowchart.

**Table 1 antibiotics-11-01440-t001:** Comparison of comorbidities between monotherapy and combination therapy groups.

	Monotherapy of Colistin Sulfate (*n* = 26)	Combined Regimen including Colistin Sulfate (*n* = 54)	t/χ^2^ Value	*p*-Value
Age(years) Average ± SD	60.5 ± 21.9	59.9 ± 19.2	0.14	0.893
Sex (Male)	19 (73.1)	40 (74.1)	0.01	0.924
Comorbidities				
Diabetes	3 (11.5)	8 (14.8)	0.16	0.690
Malignant Tumor	3 (11.5)	15 (27.8)	2.65	0.103
Cardiovascular Disease	4 (15.4)	9 (16.7)	0.66	0.884
Transplant Recipient	0 (0)	5 (9.3)	-	0.168
Respiratory Comorbidities	4 (15.4)	13 (25.0)	0.79	0.332
Chronic Liver Diseases	2 (7.7)	5 (9.3)	-	0.816
Others	5 (19.2)	11 (20.4)	0.01	0.905
History of glucocorticoids	5 (19.2)	19 (35.2)	-	0.145
History of immunosuppressants	1 (3.9)	11 (20.4)	3.76	0.053
Invasive procedures one week before Infections				
Deep-vein catheterization	14 (53.9)	19 (35.2)	2.52	0.112
Indwell gastric tube	9 (34.6)	19 (35.2)	<0.001	0.960
Indwell urethral catheter	11 (42.3)	16 (29.6)	1.26	0.261
Indwell tracheal intubation	10 (38.5)	24 (44.4)	0.26	0.612
Tracheotomy	8 (30.8)	7 (13.0)	3.65	0.056
Bronchoscopy	11 (42.3)	12 (22.2)	3.46	0.063
Operation with general anesthetics	5 (19.2)	12 (22.2)	0.09	0.759
Infectious Diseases	-	-		
Hospital-acquired pneumoniae HAP	17 (65.4)	37 (68.5)	0.08	0.779
Bloodstream infection, BSI	6 (23.1)	6 (11.1)	1.97	0.160
Acute suppurative peritonitis	0 (0)	5 (9.3)	-	0.168
Urinary tract infection	3 (11.5)	1 (1.9)	-	0.063
Acute meningitis	0 (0)	3 (5.6)	-	0.237

**Table 2 antibiotics-11-01440-t002:** Comparison of pathogenic bacteria between monotherapy and combination therapy groups.

Pathogenic Bacteria	Monotherapy of Colistin Sulfate (*n* = 26)	Combined Regimen including Colistin Sulfate (*n* = 54)	t/χ^2^ Value	*p*-Value
*Acinetobacter baumannii*	11 (42.3)	23 (42.6)	<0.001	0.981
*Pseudomonas aeruginosa*	8 (30.8)	12 (22.2)	0.68	0.408
*Klebsiella pneumoniae*	6 (23.1)	11 (20.4)	0.08	0.782
*Stenotrophomonas maltophilia*	1 (3.9)	2 (3.7)	-	0.975
*Citrobacter freudii*	0 (0)	1 (1.9)	-	0.989
≥two species of bacteria	0 (0)	5 (9.3)	-	0.168

**Table 3 antibiotics-11-01440-t003:** Clinical efficacy assessment of the monotherapy and combination therapy groups.

	Monotherapy with Colistin Sulfate (*n* = 26)	Combined Regimen including Colistin Sulfate (*n* = 54)	t/χ^2^/Z Value	*p*-Value
Time from the positive outcome of culture to the initiation of antimicrobial treatment (d) Average ± SD	2.0 ± 1.4	2.2 ± 1.4	−0.38	0.704
Treatment course (d) Average ± SD	11.7 ± 4.9	11.3 ± 4.1	0.37	0.708
Treatment course of HAP (d) Average ± SD	12.0 ± 4.8	10.6 ± 4.3	1.05	0.260
Treatment course of BSI (d) Average ± SD	9.7 ± 4.3	14.5 ± 2.1	−3.09	0.01
No. of cases in which temperature (T) returned to normal	22 (84.6)	48 (88.9)	0.29	0.588
Time taken for T cases to return to normal (d) Average ± SD	4.0 ± 3.2	5.4 ± 4.4	−1.30	0.196
Length of hospital stay (d) Average ± SD	47.2 ± 32.4	49.8 ± 35.0	−0.31	0.755
No. of cases in which WBC count returned to normal within seven days ^b^	11 (42.3)	37 (72.6)	6.71	0.001
No. of cases with decreased neutrophilic granulocyte percentage within seven days ^b^	10 (38.5)	36 (70.6)	7.39	0.007
No. of cases with decreased CRP within seven days	15 (57.7)	50 (92.6)	14.03	0.002
No. of cases with decreased PCT within seven days	14 (53.9)	47 (87.0)	10.68	0.001
Variation range of the WBC count (*10^9^/L) ^ab^ Median (Q_1_, Q_3_)	−1 (−5, 1.8)	−2.7 (−6.1, −1)	2.12	0.034
Variation range of neutrophilic granulocyte percentage ^ab^ Median (Q_1_, Q_3_)	1 (−5.5, 5.5)	−5.95 (−15, −3)	3.26	0.001
Variation range of CRP (mg/dL) ^a^ Median (Q_1_, Q_3_)	2 (−33, 16)	−21 (−54, −7)	1.96	0.051
Variation range of PCT (ng/mL) ^a^ Median (Q_1_, Q_3_)	0.0 (−0.4, 2)	−0.28 (−2.1, −0.1)	3.18	0.002
Clinical efficacy	19 (73.1)	51 (94.4)	7.33	0.007
Microbial clearance rate	13 (50.0)	40 (74.1)	4.55	0.033
28-day mortality	3 (11.5)	3 (5.6)	-	0.341
Total mortality	5 (19.2)	5 (9.3)	1.60	0.207

^a^. The index value at 7 ± 2 days—the index value at treatment initiation ±2 days. ^b^. Exclusion of three cases with decreased WBC count and neutrophilic granulocyte percentage caused by hematologic malignancy in the combined therapy group.

**Table 4 antibiotics-11-01440-t004:** Antimicrobial safety comparison between the monotherapy and combination therapy group (No. (%)).

	Monotherapy of Colistin Sulfate (*n* = 26)	Combined Regimen including Colistin Sulfate (*n* = 54)	χ^2^ Value	*p*-Value
No. of cases with elevated serum creatinine	6 (23.1)	11 (20.4)	0.08	0.782
No. of cases with elevated ALT	3 (11.5)	5 (9.3)	-	0.750
No. of cases with elevated AST	5 (19.2)	7 (13.0)	0.54	0.462
No. of cases with elevated TBil	4 (15.3)	10 (18.5)	0.12	0.730
No. of cases with decreased platelet count *	1 (4.0)	3 (5.9)	-	0.703

* Exclusion of three cases with decreased WBC count and neutrophilic granulocyte percentage caused by hematologic malignancy in the combination therapy group.

**Table 5 antibiotics-11-01440-t005:** The in vitro antimicrobial susceptibility of carbapenem-resistant Gram-negative bacteria.

	*Acinetobacter baumannii* (*n* = 23)		*Klebsiella pneumoniae* (*n* = 11)		*Pseudomonas aeruginosa* (*n* = 13)	
	MIC_50_	R%	MIC_50_	R%	MIC_50_	R%
Piperacillin/Tazobactam	≥128	100	≥128	100	≥128	92
Ceftazidime	≥64	100	≥64	100	≥64	69
Cefoperazone/Sulbactam	≥64	100	≥64	100	≥64	100
Cefepime	≥64	100	≥64	100	≥64	77
Imipenem	≥16	100	≥16	100	≥16	92
Meropenem	≥16	100	≥16	100	≥16	92
Ciprofloxacin	≥4	100	≥4	82	≥4	85
Levofloxacin	≥8	100	≥8	82	≥4	85
Amikacin	16	96	2	36	4	31
Minocycline	4	43	8	73	-	-
Doxycycline	≥16	100	≥16	91	-	-
Sulfamethoxazole/Trimethoprim	≥320	83	≥320	91	-	-
Tigecycline	1	9	2	0	-	-
Colistin	≤0.5	0	≤0.5	0	≤0.5	0

## Data Availability

The datasets used and/or analyzed during the current study are available from the corresponding author on reasonable request.

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
