# Peer review of "Combination Regimens with Colistin Sulfate versus Colistin Sulfate Monotherapy in the Treatment of Infections Caused by Carbapenem-Resistant Gram-Negative Bacilli"

_antibiotics, 2022, doi:10.3390/antibiotics11101440_

Round 1

Reviewer 1 Report

In the submitted manuscript entitled Combination regimens with colistin sulfate versus colistin sulfate monotherapy 2 in the treatment of infections caused by carbapenem-resistant gram-negative ba- 3 cilli. the authors assessed the efficacy of colistin sulfate monotherapy and combination therapy on carbapenem resistant gram negative bacilli to provide a reference for the clinical use of colistin sulfate. However, there is still scope of further improvement in the manuscript. Therefore, I recommend to accept the manuscript after a minor revision.

Here are some comments related to the manuscripts:   

1.     Improve the introduction and discuss a few latest studies related to the work.

2.     Conclusion should be more impressive and detailed.  

Although the sample size of the study is minimal but I am recommending the study for the publication in Antibiotics journal.

Reviewer 2 Report

Dear author, it is a well written manuscript.  Carbapenem-resistant gram-negative bacilli represents a worldwide problem, especially due to the lack of treatment options. This article proves that the combination therapies with colistin sulfate are more efficient in the treatment of carbapenem-resistant GNB infections over colistin sulfate alone.

I have just some suggestions:

Table no 1 -should be split in two. The comorbidities and the risks factors should be separated from the pathogenic bacteria. It is very dificult to follow.

Thank you!

Regards!

Reviewer 3 Report

This study provided an evidence based of clinical efficacy of combination of colistin with other antibiotics compared with colistin alone. Although medical recommendation/guideline suggested combination of the colistin, but this study proved the evidence based-on clinical application.

I have some comments to the author to improve their manuscript as the following;

1. Line 50-51: More details need to explain.

2. Line 54-55: What is colistin used outside China? Please mention it.

3. Line 66-69: How many sample size? It may be better if the author show overall of study by flowchart. This will make the reader to be easy understand.

4. Line 123: CLSI reference (No.7) is too old. I strongly suggested to the update version. Also please check S-I-R in Table 4 again with the update CLSI.

5. Line 126-127: What does "colistin level" meaning? Is it "resistance or intermediate"?

6. What is mortality rate between two group?

7. Please check the species name of K. pneumoniae throughout the manuscript.

8. Line 241: What is reference?

Reviewer 4 Report

The study aimed to compare the clinical efficacy and safety of colistin sulfate monotherapy and its combination with other antimicrobials in the treatment of carbapenem-resistant gram-negative bacilli (CR-GNB) infections in adults.

Here are comments for improvement of the manuscript:

Q1) Section Safety evaluation: line no. 113 “ Renal impairment was defined as  a blood creatinine level above normal or double that of the baseline “ give a normal range or define a normal range with reference

Q2)Section Safety evaluation: line no. 114 “  Liver function 114 impairment was defined as alanine aminotransferase, aspartate aminotransferase, and  total bilirubin levels above normal or double those at baseline.” give a normal range or define a normal range with reference

Q3) Line no. 105 “Microbiological efficiency was defined as the disappearance or clearance of CR-GNB from clinical specimen cultures after treatment” give references for the same.

Q4) Result Section; Line no 141 states “ The typical dosage of colistin sulfate administered to patients was 0.5 million U every 12 h……. These two regimens were only used in the monotherapy group.” It's confusing as the author is talking about two regimens and he has mentioned three. Kindly rewrite it.

Q5) Section Susceptibility testing line no 205. It's not clear from where CR-GNB strains were isolated or procured. Mention it in the method.) 

Q6) Line no. 252 “Abdelsalam reported that the clinical efficacy of” Did the author mean Abdelsalam et al 2018. Rewrite it and provide a reference.
